# State of the Art in Hepatic Dysfunction in Pregnancy

**DOI:** 10.3390/healthcare9111481

**Published:** 2021-10-31

**Authors:** Valentin Nicolae Varlas, Roxana Bohîlțea, Gina Gheorghe, Georgiana Bostan, Gabriela Anca Angelescu, Ovidiu Nicolae Penes, Roxana Georgiana Bors, Eliza Cloțea, Nicolae Bacalbasa, Camelia Cristina Diaconu

**Affiliations:** 1Department of Obstetrics and Gynecology, Filantropia Clinical Hospital, 011132 Bucharest, Romania; valentin.varlas@umfcd.ro (V.N.V.); roxana-georgiana.bors@rez.umfcd.ro (R.G.B.); eliza.clotea@gmail.com (E.C.); 2“Carol Davila” University of Medicine and Pharmacy, 020021 Bucharest, Romania; gabriela.angelescu@umfcd.ro (G.A.A.); ovidiu.penes@umfcd.ro (O.N.P.); nicolae_bacalbasa@yahoo.ro (N.B.); 3Department of Internal Medicine, “Carol Davila” University of Medicine and Pharmacy, 050474 Bucharest, Romania; gheorghe_gina2000@yahoo.com; 4Department of Gastroenterology, Clinical Emergency Hospital of Bucharest, 105402 Bucharest, Romania; 5Department of Obstetrics and Gynecology, County Emergency Hospital “St. John the New”, 720034 Suceava, Romania; bostangeorgiana@yahoo.com; 6Department of Internal Medicine, County Emergency Hospital Ilfov, 022115 Bucharest, Romania; 7Department of Anesthesiology and Intensive Care, University Clinical Hospital, “Carol Davila” University of Medicine and Pharmacy, 050474 Bucharest, Romania; 8Department of Visceral Surgery, “Fundeni” Clinical Institute, “Carol Davila” University of Medicine and Pharmacy, 050474 Bucharest, Romania; 9Department of Internal Medicine, Clinical Emergency Hospital of Bucharest, 105402 Bucharest, Romania

**Keywords:** pregnancy, liver dysfunction, hyperemesis gravidarum, HELLP syndrome, acute fatty liver, intrahepatic cholestasis, cholelithiasis, Budd–Chiari syndrome, cirrhosis

## Abstract

Hepatic dysfunction in pregnant women is always challenging for the obstetrician, as the spectrum of hepatic abnormalities can be very large and have various implications, both for mother and fetus. There is a diagnostic and therapeutic polymorphism of hepatic dysfunction in pregnancy and insufficient knowledge related to the etiopathogenesis and epidemiology of this disease. The clinical forms of hepatic dysfunction encountered in pregnancy can vary from liver diseases related to pregnancy (e.g., HELLP syndrome, intrahepatic cholestasis, hyperemesis gravidarum, or acute fatty liver of pregnancy) to de novo ones occurring in pregnancy, and pre-existing liver disease (cholelithiasis, Budd–Chiari syndrome, and cirrhosis). We performed a systematic literature search over 10 years. The review protocol assumed a search of two databases (PubMed^®^/MEDLINE and Web of Science Core Collection). The strategy regarding the management of these diseases involves multidisciplinary teams composed of different specialists (obstetricians, gastroenterologists and anesthetists) from specialized tertiary centers. Despite the improving prognosis of pregnant women with liver diseases, the risk of maternal–fetal complications remains very high. Therefore, it is necessary to ensure careful monitoring by a multidisciplinary team and to inform the patients of the potential risks.

## 1. Introduction

Hepatic dysfunction in pregnant women is always challenging for obstetricians, as the spectrum of hepatic abnormalities is very large and can have various implications both for mother and fetus. The therapeutic decision must consider the liver’s physiologic changes during pregnancy and the severity of hepatic impairment. Hepatic dysfunction is found in pregnancy with a prevalence of about 3% [1,2]. Severe liver dysfunctions in pregnancy are associated with an increased rate of maternal–fetal morbidity and mortality. The overall incidence of maternal death from liver disease in pregnancy is around 6%, and most studies reveal a specific mortality rate for each condition [2,3,4]. Casey et al. reported in a study including 3155 patients from 33 tertiary care liver centers over the course of 20 years, that half of the acute liver failures during pregnancy were associated with hemolysis, elevated liver enzymes and low platelets (HELLP) syndrome, and acute fatty liver of pregnancy (AFLP). Furthermore, delivery and intensive supportive measures were performed in most cases; 16% required liver transplantation, and 11% deceased [4].

Liver diseases are classified according to their relationship with pregnancy: some are specific to pregnancy (e.g., intrahepatic cholestasis, AFLP, or systemic diseases with hepatic manifestations, such as preeclampsia), while others represent nonspecific diseases that can also occur in nonpregnant women (e.g., acute viral hepatitis) (Table 1). In clinical practice, the incidence of pregnancy-related liver dysfunction increased progressively from acute fatty liver in pregnancy to pre-eclamptic liver dysfunction (Figure 1) [5,6]. A significant factor for the overall prognosis is liver function before pregnancy. Some conditions may worsen during pregnancy, such as cholelithiasis, Budd–Chiari syndrome, and cirrhosis [7] (Table 1). The clinician must know this fact for appropriate counseling. There are also some physiologic, anatomic, and functional changes of the liver during pregnancy. Palmar erythema and spider angiomas can be caused by the hyperestrogenemia of pregnancy and are not necessarily signs of chronic liver disease. The ultrasound examination usually shows a normal liver and biliary tract. However, the gallbladder volume can be increased during pregnancy. Laboratory liver function tests reveal physiologic changes, such as increased serum alkaline phosphatase levels, 5’-nucleotidase, slightly decreased serum albumin level, gamma-glutamyl transpeptidase, and total and unconjugated bilirubin. Aminotransferases and total bile acid concentrations remain within normal ranges [7].

Pregnancy-related liver disease is dependent on the trimester, as opposed to those which are not pregnancy-related, which can occur at any time, independent of the trimester of pregnancy. The establishment of the diagnosis and therapeutic management depends on the moment of the symptoms’ onset and alterations of laboratory tests [8].

Special attention should be paid to the liver dysfunction associated with preeclampsia, mainly due to the increased incidence during pregnancy compared to other liver diseases related or unrelated to pregnancy. Moreover, this pathological condition represents a continuous challenge, as the incidence of hypertension is increasing.

Preeclampsia is a complication that occurs after 20 weeks of pregnancy and is characterized by hypertension (blood pressure > 140/90 mm Hg) and proteinuria, with or without edema. It complicates 3% of all pregnancies and causes liver dysfunction in about half of cases, suggesting a severe form of the disease [9]. Hepatic microcirculatory disorders and hepatocellular necrosis occur because of vascular endothelial damage. In mild forms of pre-eclampsia, a 1.5–5 times increase in the serum levels of aminotransferases and alkaline phosphatase is identified. In severe forms, thrombocytopenia can appear in cases complicated with disseminated intravascular coagulation (DIC), and hemolytic jaundice may be present, with total serum bilirubin around 6 mg/dL [1,2]. Pregnant women may be asymptomatic or associate upper abdominal pain, nausea, vomiting, headache, or visual symptoms. Preeclampsia increases the risk of HELLP syndrome, but not all pregnant women with HELLP have preeclampsia [2].

To manage preeclamptic liver dysfunction and prevent further complications (liver rupture, hepatic infarction, convulsions, and acute renal failure), it is necessary to urgently administer antihypertensive treatment to control the blood pressure. In addition, magnesium sulfate is used to reduce the risk of seizures (eclampsia). In mild forms of preeclampsia, birth is recommended after 36 weeks to reduce the incidence of serious complications [10,11]. Prevention of severe preeclampsia is performed by administering aspirin at ≤16 weeks of gestation [9].

The research from recent years has led to an improvement in the maternal–fetal prognosis among pregnant patients with liver disease by improving the diagnostic and therapeutic management. An important step is represented by the approach of these patients by a multidisciplinary team, consisting of gynecologists, obstetricians, and gastroenterologists. The maternal–fetal prognosis is given primarily by the type of liver disease, the degree of impairment of liver function, and the adaptation of treatment according to gestational age [12,13]. One challenge is to correctly identify the liver changes that occur physiologically in pregnancy [14]. It is important for women of child-bearing age with pre-existing liver disease to be informed of the risks involved in pregnancy [7].

This article reviews the diagnosis and most important therapeutic decisions a clinician should take in hepatic dysfunction during pregnancy.

## 2. Materials and Methods

### 2.1. Search Strategy

The authors performed a systematic literature search according to the 2020 PRISMA checklist, on the following MESH terms: (1) “HELLP syndrome,” (2) “acute fatty liver,” (3) “intrahepatic cholestasis,” (4) “cholelithiasis,” (5) “Budd-Chiari,” (6) “cirrhosis,” (7) “hyperemesis gravidarum”, with a data filter on (AND) “pregnancy” searching over the course of 10 years. In the present paper, we focused the search with filters: clinical trial (CT) and randomized controlled trial (RCT); the last updated search was performed on 18 July 2021. The review protocol assumed a search on PubMed^®^/MEDLINE and Web of Science Core Collection.

### 2.2. Study Selection

Inclusion criteria were the following: clinical (CT) or randomized controlled trials (RCT), published in the last 10 years, investigating (1) biomarkers of ICP among studies, (2) biomarkers of HELLP syndrome, (3) treatment options in the most frequent dysfunctions of the liver in pregnancy, and (4) main findings in hepatic dysfunction in pregnancy.

Exclusion criteria were the following: (1) other types of documents than RCT and CT, (2) articles with insufficient information, (3) studies published before 2011, and (4) studies investigating other organ dysfunction during pregnancy.

### 2.3. Data Extraction and Synthesis

To carry out the study, data were extracted and analyzed regarding the year of publication, the language in which the study was written, the relevance and the accuracy of the information. We analyzed the data following the study objectives. The statistical analysis was completed using Microsoft Excel^®^ 2013 (Microsoft^®^ Corporation, Redmond, WA, USA).

The PubMed^®^/MEDLINE data search was for RCT and CT in humans during the last ten years, using the selected keywords: (1) “HELLP syndrome”, (2) “acute fatty liver”, (3) “intrahepatic cholestasis”, (4) “cholelithiasis”, (5) “Budd–Chiari” (6) “cirrhosis”, (7) “hyperemesis gravidarum”, with a data filter on (AND) “pregnancy”. Results: 147 published papers.

We searched for TOPIC ((pregnancy) AND ((HELLP syndrome) OR (acute fatty liver) OR (intrahepatic cholestasis) OR (cholelithiasis) OR (Budd–Chiari) OR (cirrhosis) OR (hyperemesis gravidarum)). Timespan: 2011–2021. Databases: SCI-EXPANDED, SSCI, BKCI-S, and ESCI. Results: 145 published papers.

## 3. Results

More than 292 reports were screened for eligibility according to the topic search. 22 papers were identified, analyzed, and included in the review (Figure 2).

### Study Characteristics

Table 2, Table 3 and Table 4 provide an overview of 22 of the included studies. Table 2 includes 2 studies, both of which are CCS, which investigated biomarkers of HELLP syndrome. Table 3 includes six studies, all of them being case-control studies (CCS), which investigated biomarkers of ICP. Table 4 includes 14 studies that investigated the outcome of a particular treatment. A total of six studies were randomized controlled trials (RCT), a semi-factorial RCT, three case-control series, two prospective studies, and two retrospective multicentric studies. Of the studies, four were focused on the topic of HELLP syndrome, nine were focused on ICP, and one study focused on AFLP. Table 5 summarizes the main characteristics for each hepatic dysfunction in pregnancy, containing the essential information about the epidemiology, clinical and paraclinical findings, therapeutic management, prognosis, and fetal outcome.

## 4. Hyperemesis Gravidarum and Liver Dysfunction

Hyperemesis gravidarum (HG) is a complication of early pregnancy that occurs in 0.3–2% of all pregnancies, characterized by excessive vomiting with electrolyte imbalance, ketosis, or loss of more than 5% of body weight during pregnancy [37,38]. In addition to dyselectrolytemia, kidney and thyroid dysfunction, liver function can also be impaired without clear pathogenesis. In pregnant women with HG, the appearance of jaundice suggests liver damage [38]. Changes in liver function occur in 50% of pregnant women with HG who require hospitalization [8]. The severity of nausea and vomiting in patients with HG and liver disease correlates with the increase in liver enzymes [7].

Alanine aminotransferase (ALT) is the most commonly impaired enzyme, it increases one to three times the upper limit of normal to 200 U/L, rarely exceeding 1000 U/L. ALT increases more than AST [1]. Alkaline phosphatase may double, and bilirubin increases up to 4 mg/dL. Synthetic liver function remains intact, without affecting the coagulation profile and serum albumin levels, except in cases in which HG has been complicated by malnutrition. The liver function returns to normal within a few days after cessation of vomiting [6]. HG with hepatic dysfunction does not leave long-term sequelae and is not fatal, with no reported cases of fulminant hepatic failure [1].

HG can promote biliary sludge formation due to dehydration, condensation of the bile, and starvation with Oddi’s sphincter contraction. The presence of sludge increases the possibility of jaundice appearance, secondary to cholecystitis [39].

Regarding management, hospitalization is necessary to correct possible hydroelectrolytic imbalances and combat vomiting by administering intravenous fluids and antiemetics, intestinal rest, and, rarely, parenteral nutrition [38].

The effectiveness of pharmacological therapy depends on initiating antiemetic treatment to reduce the severity of nausea and vomiting. After the onset of symptoms, vitamin B6 (pyridoxine) or the combination of doxylamine and vitamin B6 may be useful, as first-line pharmacotherapy [40].

When oral therapy cannot be tolerated, the following routes may be used: intravenously, intramuscularly, rectally (promethazine and prochlorperazine suppositories), or transdermal (granisetron patches). Continuous subcutaneous pump microinfusion of metoclopramide or ondansetron is not used due to adverse effects [41,42].

## 5. HELLP Syndrome

HELLP syndrome is an acronym used to define the association between hemolysis with a microangiopathic blood smear (defined by the presence of schistocytes), elevated liver enzymes (serum AST > 2 times upper limit of normal), and a low platelet count (≤100,000 cells/microL). HELLP syndrome may occur in 0.1–0.6% of all pregnancies, late in the second trimester (25–38 weeks) in 70% of cases, and early postpartum in 30% of cases [1].

As a result of hemolysis, serum indirect and total bilirubin increase, and serum haptoglobin decreases (≤25 mg/dL). It can be related to preeclampsia, but in 15–20% of cases, there is no personal history of hypertension or proteinuria. Both severe preeclampsia and HELLP syndrome can determine severe hepatic lesions, such as infarction, hemorrhage, and rupture [7,43].

There are several risk factors for HELLP syndrome, such as the personal history of preeclampsia or HELLP syndrome, or multiparity. The signs and symptoms of HELLP syndrome are less specific and can be easily mistaken for gastroenteritis, gallbladder disease, or viral hepatitis. The usual symptoms of HELLP syndrome include abdominal pain in the epigastrium or right upper quadrant, nausea, and vomiting. Hypertension (blood pressure ≥140/90 mmHg) is present in almost 85% of cases, but the absence of hypertension does not exclude the presence of HELLP syndrome. Less frequently, the patient presents with headaches, visual changes, jaundice, and ascites. The symptoms usually develop during the last trimester of pregnancy, but the onset of the disease can occur during the second trimester of pregnancy or the postpartum period, within 48 h of delivery [44,45].

According to the Mississippi and Tennessee classifications, predictable diagnostic criteria are used to manage HELLP syndrome [7,46,47] (Table 6).

For the early diagnosis of HELLP syndrome, some studies suggest using novel genetic biomarkers in association with laboratory markers, new potential tools with clinical utility (Galectin-1 and p65/RelA immunoexpression of placental NF-kB) [15,16] (Table 2).

Several diseases must be differentiated from HELLP syndrome, such as AFLP, acute hepatitis, gastroenteritis, appendicitis, gallbladder dysfunction, lupus flare, antiphospholipid syndrome, hemolytic-uremic syndrome (HUS), immune thrombocytopenia (ITP), or thrombotic thrombocytopenic purpura (TTP), and nonalcoholic fatty liver disease (NAFLD). Sometimes the differential diagnosis can be very difficult. In the case of AFLP, the clinical presentation can be similar. The presence of coagulopathy, severe hypoglycemia, and elevated serum creatinine are more frequent in AFLP than HELLP syndrome. Hypertension is more common in HELLP syndrome than in AFLP [47]. In the case of thrombotic microangiopathy, the laboratory findings are suggestive: severe thrombocytopenia and anemia, elevated serum lactate dehydrogenase (LDH), with a minimal elevation of AST, and a higher percentage of schistocytes on a peripheral blood smear (2–5% in TTP compared to less than 1% in HELLP syndrome). The onset of the disease is also important. TTP tends to occur earlier than HELLP syndrome during gestation (12% during the first trimester, 56% during the second trimester, and 33% in the third trimester/postpartum). Severe renal insufficiency is more frequently associated with the hemolytic uremic syndrome (HUS) than HELLP syndrome [8,49].

HELLP syndrome is considered an obstetric emergency. The therapeutic decisions must consider the gestational age and status of the mother and fetus. Delivery is regarded as the only effective treatment [50]. First of all, the mother should be stabilized using supporting measures: treatment of hypertension with labetalol, nifedipine, urapidil or hydralazine, prophylaxis of seizures with magnesium sulfate in patients with severe preeclampsia, and platelet transfusion in the case of severe thrombocytopenia with active bleeding or a very low platelet count (less than 20,000 cells/microL). Immediate delivery after maternal stabilization is indicated in the following settings: gestational age ≥ 34 weeks or < 23 weeks, fetal demise, non-reassuring tests of fetal status (biophysical profile and fetal heart rate testing); or severe maternal conditions: multiorgan dysfunction, consumption coagulopathy, liver infarction or hemorrhage, pulmonary edema, renal failure, or abruptio placenta [1]. In the case of pregnancies ≥ 23 and <34 weeks of gestation with reassuring maternal and fetal status, a short course of corticosteroids before delivery is considered safe. However, delivery should not take place beyond 48 h from the onset of the disease. The usual obstetrical indicators decide the route of delivery. Vaginal delivery should be attempted when possible (favorable cervix for induction of labor, a vertex presenting infant, gestational age over 30–32 weeks). Both cesarean section and vaginal delivery are safe when platelet count > 50000/microL [51,52,53].

Hepatic dysfunction in HELLP syndrome is a separate entity that requires special management. Very high levels of aminotransferases should raise the suspicion of hepatic infarction, subcapsular hematoma caused by hepatic rupture, or an unrelated etiology, such as viral hepatitis. Hepatic infarction and subcapsular hematoma are associated with sudden, severe right upper quadrant abdominal pain, shoulder and neck pain, or hypotension. Subcapsular hematoma may remain contained or may cause hemoperitoneum and shock [11].

Computed tomography (CT) or magnetic resonance imaging (MRI) is useful to exclude the hepatic complications of HELLP syndrome. The management of a contained hematoma is usually conservative, with blood transfusion and volume replacement in some cases. Emesis and cough should be avoided, as they are associated with increased intraabdominal pressure. Several imaging studies and laboratory tests should be performed to observe if hematoma remains stable over time and aminotransferases decrease. The complete resolution of the hematoma will probably occur in a few months, and the patient can be discharged. An alternative to conservative management for hemodynamically stable patients is percutaneous embolization of the hepatic arteries [50].

Surgical intervention is required in case of hemodynamic instability or persistent expansion of the hematoma. Surgical techniques include drainage of the hematoma and packing, hepatic artery ligation, and partial resection of the liver. Liver transplantation is mentioned in some case reports [50,54,55,56].

Fetal outcome and long-term prognosis depend especially on the gestational age at delivery and birth weight. The prematurity rate among these infants is very high (70%), and approximately 15% of births occur before 27 weeks. The fetal outcome may also be impaired by the presence of placental abruption or fetal growth restriction. Studies report a perinatal mortality rate of 7–20% [57,58].

Recovery after HELLP syndrome can take weeks, depending on the severity of clinical presentation and organ dysfunction. Laboratory tests may worsen during the first 48 h after delivery. Recovery may take more time in women with DIC, platelet count <20,000 cells/microL, renal failure, or hepatic complications. Maternal symptoms usually improve within 48 h after delivery. Women with HELLP syndrome have an increased risk of future complications when they become pregnant again (7% risk of HELLP syndrome, 18% risk of preeclampsia, and a 20% risk of gestational hypertension). HELLP syndrome is not considered a cause of long-term renal dysfunction. Unfortunately, no therapy can prevent the development of HELLP syndrome during pregnancy [59,60].

Although there is no clear evidence of the benefit of corticosteroids on clinical outcomes, they appear to be effective in HELLP class I syndrome for accelerating postpartum recovery [23,26]. In the case of liver rupture, maternal and fetal mortality rates are estimated to be approximately 50%, and the main causes of death are blood loss and coagulation disorders. Patients may also suffer from acute respiratory distress syndrome (ARDS), pulmonary edema, and acute liver failure [54].

## 6. Acute Fatty Liver of Pregnancy

AFLP is a rare disorder, with an approximate incidence of 1 in 7000 to 1 in 20,000 deliveries [61,62,63]. Other authors showed an incidence of about 0.05% of all pregnancies [62]. The microvesicular fatty infiltration of hepatocytes characterizes it. This disease’s pathogenesis is incompletely known, but the most common factor involved is a defect in mitochondrial beta-oxidation of fatty acids (long-chain 3-hydroxyacyl CoA dehydrogenase deficiency). The disease is more frequently observed in the case of multiple pregnancies and in underweight women. AFLP usually develops during the last trimester of pregnancy and is considered an obstetric emergency [64].

The signs and symptoms are almost the same as those of HELLP syndrome and include nausea or vomiting, abdominal pain in the epigastric region or upper right quadrant, malaise, anorexia, and jaundice. Approximately 50% of patients have preeclampsia. A particular feature of AFLP is the development of central diabetes insipidus (with polyuria and polydipsia), since the clearance of vasopressinase is impaired in severe hepatic dysfunction [65].

The diagnosis of AFLP is based on clinical findings correlated with laboratory data. Considering the similarities with HELLP syndrome, it is sometimes difficult to make a differential diagnosis [54] (Table 7).

The criteria for the diagnosis of AFLP include the presence of a minimum of six of the following parameters, in the absence of another explanation: clinical findings compatible with the diagnosis (vomiting, abdominal pain, polydipsia/polyuria, and encephalopathy), laboratory findings such as elevated bilirubin (>14 μmol/L), leukocytosis (>11 × 109/L), elevated uric acid (>340 μmol/L), hypoglycemia (<4 mmol/L), the elevation of serum aminotransferases (>42 IU/L), renal dysfunction (creatinine > 150 μmol/L), elevated ammonia (>47 μmol/L), coagulation disorder (prothrombin time > 14 sec or activated partial thromboplastin time > 34 sec), the presence of ascites on ultrasound scan, and microvesicular steatosis on liver biopsy. Compared to HELLP syndrome, bilirubin levels are high, while aminotransferases are only moderately increased [65,66]. Hypoglycemia is a characteristic feature caused by hepatic insufficiency and can be useful for differential diagnosis [7].

Imaging studies of the liver are not very reliable for the diagnosis of this pathology. A study that compared ultrasound with MRI and CT revealed that the best modality to reveal the presence of fat in the liver was CT. However, it was successful in only 50% of all cases [54]. Imaging studies are useful to exclude other possible complications, such as liver infarction of hematoma. Liver biopsy is a useful diagnostic tool, rarely used because it is an invasive procedure, the diagnosis requires time, and the patient needs emergency treatment [67].

Treatment involves prompt delivery, regardless of gestational age and maternal stabilization. Delivery should take place in the first 24 h after diagnosis. The delivery route is decided by considering maternal and fetal status, gestational age, the presenting part of the infant, and the possibility of successful labor induction. Cesarean section is indicated in all cases when rapid delivery of the baby cannot be accomplished. Maternal stabilization includes correction of hypoglycemia with glucose infusion, treatment of hypertension, and coagulation disorders (administration of fresh frozen plasma, cryoprecipitate, packed red blood cells, and platelets). Laboratory analysis should be conducted every several hours, as maternal status can change rapidly over time. These patients have an increased risk of bleeding [68,69]. Early initiation of plasmapheresis in pregnancy or its rapid onset after birth improves maternal outcome [70].

AFLP is a complex disease. Most patients have a recovery of liver function without sequelae. However, many studies report high mortality rates for both mother and fetus because of multiorgan dysfunction at the moment of diagnosis. Potential complications of AFLP include renal dysfunction and pancreatitis. Recent data show that prompt diagnosis and treatment led to decreased maternal mortality rate from 75% to 18% and perinatal mortality rate from 85% to 23%. The fetal impaired outcome is due to prematurity and low birth weight. The recurrence rate of AFLP in subsequent pregnancies is not well-known. The recurrence risk in subsequent pregnancies is higher for a mother carrying a long-chain 3-hydroxyacyl-coA dehydrogenase (LCHAD) gene mutation. However, LCHAD deficiency is uncommon, and only a few cases were documented. Women should be aware that recurrence can occur and should be closely monitored for during subsequent pregnancies [68].

## 7. Intrahepatic Cholestasis of Pregnancy

Intrahepatic cholestasis of pregnancy (ICP) is a disease that typically develops during the late second or third trimester of pregnancy and remission after delivery [71]. Depending on ethnicity, geographical location, and seasonal pattern, the incidence varies between 0.2–2% [71,72]. It is the most common liver disease during pregnancy and results from the interaction between genetic, hormonal, and environmental factors. ICP is an inflammatory condition with a possible maternal immune mechanism [18]. The disease pathogenesis is not entirely understood, but it demonstrated a familial clustering with an increased risk in first-degree relatives and some ethnic groups. In the diagnosis of ICP and laboratory markers, a series of new genetic markers have been proposed, with potential clinical utility. In ICP, some biomarkers are used for a correct diagnosis (long noncoding RNA, neopterin, maternal urinary mRNAs, and native thiol) and others in the evaluation of the treatment (CRH) [17,18,19,20,21,22] (Table 3).

A series of changes in the genes BSEP/ABCB11, MDR3/ABCB4, GABRA2, and ATP8B1/FIC1 were observed in patients with ICP, which is associated with lesions of biliary ducts and hepatocyte membrane. Furthermore, severe ICP forms show genetic variants of MDR3 and BVB [73,74,75].

A personal history of ICP is associated with a high risk of recurrence in future pregnancies (approximately 60–70%). Pregnancy is associated with high serum levels of estrogen and progesterone, which alter bile metabolism and are a risk factor for cholestasis. ICP was reported more frequently in women with ovarian hyperstimulation, multiple pregnancies, or those previously treated with oral contraceptives. The disease more frequently develops in women with underlying liver diseases such as hepatitis C or nonalcoholic cirrhosis, but in most cases, there is no personal history of liver dysfunction [71,73].

Patients usually present with pruritus located on the palms and soles, and scratch marks. In some cases, the patient suffers from right upper quadrant abdominal pain, nausea, loss of appetite, sleep deprivation, or steatorrhea. Jaundice occurs in only 14–25% of cases and develops one to four weeks after the onset of pruritus [1,71].

The diagnosis is based on clinical and laboratory data. Laboratory findings include a high serum level of total bile acid, a slight increase in total and direct bilirubin (total bilirubin levels are usually less than 6 mg/dL), and a slight increase in serum aminotransferases and gamma-glutamyl transpeptidase. The prothrombin time may be prolonged in severe steatorrhea with malabsorption of vitamin K. Severe cholestasis accounts for 20% of all cases and is defined as bile acids concentration over 40 micromol/L. The ultrasound evaluation of the liver and biliary tract does not show abnormalities [72,76].

ICP can be associated with poor fetal outcome if the disease is not recognized and treated. Maternal bile acids cross the placenta and concentrate in fetal tissues, causing an increased risk of fetal stillbirth, meconium-stained amniotic fluid, preterm delivery, and neonatal respiratory distress syndrome. Fetal demise may be related to the development of arrhythmia caused by bile acids or the development of placental vasospasm. Bile acids may increase the expression of myometrial oxytocin receptors and cause preterm birth [77]. Studies report rates of spontaneous preterm birth of 30–40% in women with intrahepatic cholestasis and a rate of meconium staining of amniotic fluid as high as 16–58% [78].

The risk of fetal complications is related to the serum concentration of bile acids (the risk is increased by 1–2% for each additional μmol/L of bile acid). The risk of poor fetal outcome is higher when bile acid levels are over 40 μmol/L, and remains similar to a normal pregnancy when bile acid levels are less than 40 μmol/L [79,80].

A retrospective study was conducted in the Netherlands on a total of 215 women with intrahepatic cholestasis. Severe intrahepatic cholestasis was defined as a serum concentration of bile acids of more than ≥100 μmol/L. Authors report that gestational age at diagnosis and gestational age at delivery was lower in severe intrahepatic cholestasis than in cases with mild intrahepatic cholestasis. The rates of preterm birth (19.0%), meconium-stained fluid (47.6%), and perinatal death (9.5%) were higher in cases with severe intrahepatic cholestasis. Maternal bile acid levels at diagnosis and at delivery were positively correlated with umbilical cord blood bile acid levels [81].

Maternal treatment has two main goals: to ameliorate symptoms and reduce the risk of perinatal morbidity and mortality. Treatment includes administering ursodeoxycholic acid (UDCA) in a dose of 15 mg/kg per day, which reduces serum concentration of bile acids and pruritus within one or two weeks [29,82]. The drug proved useful in reducing premature birth, respiratory distress syndrome, and neonatal intensive care unit admission. Most cases respond favorably to the therapy. Chappell et al., in a double-blind, multicenter, randomized, placebo-controlled trial, reconsidered the treatment with UDCA in women with ICP, mentioning that it does not reduce adverse perinatal results [34,83].

Some refractory cases may benefit from treatment with S-adenosyl-methionine or cholestyramine. Cholestyramine is given in a dose of 2 to 4 g per day and is gradually increased (the maximum amount is 16 g per day). The drug may cause constipation and malabsorption of fat-soluble vitamins, such as vitamin K, especially in doses higher than 4 g per day. Considering the risk of vitamin K deficiency, oral vitamin K is recommended in a dose of 10 mg per day, from the moment of diagnosis until delivery [82,84].

The biophysical profile score is less useful to predict fetal outcomes, as the mechanism of death is not related to placental insufficiency. The most recent guidelines recommend delivery at 37 weeks of gestation because the risk of fetal demise progressively increases [72]. The delivery route is not influenced by the presence of this disease and should be dictated only by obstetrical factors. Delivery before 37 weeks of gestation may be considered in the following cases: unremitting maternal pruritus with medication, jaundice, a prior history of fetal demise before 36 weeks due to intrahepatic cholestasis, serum bile acid concentration ≥100 micromol/L. The maternal outcome is generally good, and the symptoms generally disappear after delivery. Laboratory tests should be repeated six to eight weeks after delivery to confirm that liver function is within normal limits [85].

## 8. Cholelithiasis

Pregnancy and postpartum are associated with an increased risk of developing biliary sludge and gallstones [86] because the bile increases cholesterol saturation and equally decreases gallbladder motility [87]. The pathophysiological changes in the biliary system that favor the appearance of gallstones are secondary to the hormonal changes in pregnancy. The increase in bile cholesterol saturation is not only due to the increase in estrogen-induced cholesterol secretion but also to the decrease in progesterone-induced bile acid secretion. In addition, progesterone reduces gallbladder motility, thus favoring biliary stasis and the formation of stones [88]. Changes in the biliary system disappear in postpartum, with the possibility of the remission of biliary sludge or gallstones [89,90]. However, the probability of resolution is higher for sludge than gallstones (39% vs. 9%) [90].

The incidence of asymptomatic gallstones varies between 5 and 12%, and the incidence of gallbladder disease in pregnancy is between 0.05 and 0.3% [91]. Between 2% and 31% of pregnant women may develop biliary stones or sludge. The incidence of acute cholecystitis varies between 1–8 cases per 10,000 pregnancies [92]. Gallstone disease is the most common non-obstetrical cause of hospitalization in the postpartum period [92]. Obesity, advanced maternal age, multiparity, predisposing genetic background, and a history of biliary disease before pregnancy are among the risk factors for developing gallstones during pregnancy [90,93]. A study of 3070 pregnant women, that examined the correlation between the formation of gallstones and dietary fat, fat subtypes, dietary protein, and protein subtypes, revealed the absence of a causal relationship between them. The same study suggests a potential role of carbohydrate intake or hyperinsulinemia in the development of gallstones in pregnancy [92].

The diagnosis of cholelithiasis in pregnant women is similar to the diagnosis in the antepartum period and is based on clinical and paraclinical evaluation (Table 8). Clinically, the presentation varies from the complete absence of symptoms to the presence of typical biliary symptoms such as abdominal pain, vomiting, nausea, anorexia or atypical symptoms such as belching, regurgitation, early satiety, abdominal distension, nausea or vomiting alone, epigastric or retrosternal burning, chest pain or nonspecific abdominal pain [94].

The pain usually occurs 1–3 h postprandially and is preceded by ingestion of fatty foods. The intensity of the pain varies from a slight discomfort in the epigastrium or right hypochondrium to a very intense, excruciating pain, lasting more than an hour. Rarely, pregnant women may present with symptoms of acute cholecystitis, such as abdominal pain, fever, anorexia, nausea, and vomiting. In this case, the pain is constant, intense, and long-lasting (between 4 and 6 h) [94,95]. Some patients may have atypical symptoms, such as belching, regurgitation, early satiety, abdominal distension, nausea or vomiting alone, epigastric or retrosternal burning, chest pain, or nonspecific abdominal pain, in which case there is a need for paraclinical investigations [96]. These include both biological and imaging tests. Laboratory tests are usually normal in the absence of complications. The appearance of leukocytosis, increased liver function tests (aminotransferases, bilirubin and alkaline phosphatase), or pancreatic tests (amylase and lipase) can indicate the development of a complication, such as cholecystitis, cholangitis, or pancreatitis [97]. Laboratory tests are useful for the differential diagnosis with other conditions that may have a clinical presentation similar to that given by the presence of gallstones. Figure 3 highlights the main conditions that require differential diagnosis with cholelithiasis [91].

For the diagnosis of gallstones in pregnant women, ultrasonography, MRI, cholescintigraphy with 99mTc-hepatic iminodiacetic acid (HIDA) scan, or endoscopic retrograde cholangiopancreatography (ERCP) may be used [98,99,100]. CT scans or X-rays are usually avoided in pregnant women because of the risk of exposing the fetus to ionizing radiation. Their effectiveness is also inferior to that of ultrasound or MRI [98].

The most effective and safe method that can identify gallstones in pregnant women is transabdominal ultrasonography. Its sensitivity in identifying gallbladder stones varies between 85–95%, and its specificity is about 95% [99]. The presence of signs such as ultrasonographic Murphy’s sign, gallbladder distension, pericholecystic fluid, and gallbladder wall thickening can indicate the presence of cholecystitis. However, the efficiency of this imaging method is low for the identification of gallstones, and performing an ERCP is preferable. Magnetic resonance cholangiopancreatography (MRCP) may be helpful when some complications, such as choledocholithiasis or pancreatitis, are suspected, if transabdominal ultrasonography does not provide sufficient information. Although no harmful effects of MRI on the fetus are known, MRI is not recommended in the first trimester of pregnancy. This is due to uncertainty about the effects of MRI on fetal safety during organogenesis. Furthermore, the administration of gadolinium in pregnant women is controversial [98]. A hepatobiliary iminodiacetic acid (HIDA) scan is rarely recommended. A dose of <5 milligrays did not increase the risk of fetal abnormalities [100].

There are four possible scenarios of cholelithiasis in pregnancy (Figure 4):Asymptomatic women, in whom gallstones are discovered incidentally on ultrasound examination.Women with typical biliary symptoms and the presence of gallstones on ultrasound examination.Women with atypical biliary symptoms and the presence of gallstones on ultrasound examination.Women with typical biliary symptoms and the absence of gallstones on ultrasound examination [91,92,93,94].

The management of biliary colic in pregnant women includes symptomatic, supportive, and antibiotic treatment in selected cases (acute cholecystitis or cholangitis) or even surgical treatment. Cholecystectomy can be performed safely in any trimester of pregnancy, and the laparoscopic approach is preferable [101,102]. For pregnant women who develop cholecystitis in the first two trimesters of pregnancy, cholecystectomy is recommended. Surgical excision of the gallbladder in the third trimester increases the risk of premature birth. If the patient does develop biliary colic or an episode of cholecystitis in the third trimester, the approach is to try to delay the surgical intervention. If signs of sepsis, gangrene or perforation occur, emergency intervention involving cholecystectomy, gallbladder drainage or biliary tract drainage is recommended [101,102].

## 9. Budd–Chiari Syndrome

Budd–Chiari syndrome (BCS) is a condition characterized by obstruction of hepatic venous flow, leading to sinusoidal congestion with liver dysfunction, ischemic liver injury, and portal hypertension [103]. The incidence of BCS varies significantly among studies, between 0.16–4.1 per million [104]. Depending on the origin of the obstructive lesion, BCS is classified into primary and secondary. When the obstruction is due to an endoluminal obstruction, such as thrombosis, the BCS is considered primary, while extrinsic compression or tumor invasion of the blood vessel leads to secondary BCS [105]. Usually, patients with BCS have one or more thromboembolic risk factors [103]. Murad et al. reported in a study that followed 163 patients with BCS that 84% of these patients had at least one thromboembolic risk factor, and 46% of patients had more thromboembolic risk factors [106]. The risk factors for BCS include:Thrombophilia: factor V Leiden G1691A mutation, prothrombin gene G20210A mutation, protein C deficiency, protein S deficiency, antithrombin deficiency, antiphospholipid antibodies syndrome, hyperhomocysteinemia, and paroxysmal nocturnal hemoglobinuria.JAK2V617F mutation.Myeloproliferative disorders, such as polycythemia vera, essential thrombocythemia, and idiopathic myelofibrosis.Hormonal factors: oral contraceptive use, pregnancy.Connective tissue disease: inflammatory bowel disease, Behçet disease, sarcoidosis, and vasculitis.Dehydration [105].

The most common risk factor for BCS is a myeloproliferative neoplasm, a condition encountered in approximately 49% of patients with BCS [103,106]. Another very important risk factor is hormonal changes. Thus, women under treatment with oral contraceptives, pregnant women, or women in the first two months postpartum represent approximately 20% of patients with hepatic venous outflow obstruction [107].

The increased risk of BCS in pregnancy is due to hypercoagulability, a physiological mechanism favoring childbirth [108]. Moreover, pregnancy can exacerbate other procoagulant disorders, such as protein S deficiency, factor V Leiden G1691A mutation, antiphospholipid antibodies syndrome, or nocturnal paroxysmal hemoglobinuria [107,108].

The diagnosis of BCS in pregnant women is similar to the diagnosis in the antepartum period and is based on clinical and paraclinical evaluation (Table 9) [109].

The management of patients with BCS includes many options. First, treatment of the underlying disease leading to the procoagulant status must be initiated. The patient may undergo anticoagulant treatment in the absence of contraindications and treatment of portal hypertension. There may be a need to perform thrombolytic treatment, angioplasty, or stenting for symptomatic patients undergoing angiographic treatment. Patients with acute liver failure or whose initial treatment has failed may need liver transplantation, transjugular intrahepatic portosystemic shunt (TIPS) placement, or surgical shunting [108].

The therapeutic management of BCS in pregnancy is similar to that of BCS management outside of pregnancy, albeit with some specifications [107]:Anticoagulant treatment with vitamin K antagonists is not recommended in pregnancy.Pregnancy and childbirth can be performed safely if the patient receives appropriate supportive and anticoagulant treatment.Liver transplantation can be used as a life-saving therapeutic method [110].

The prognosis of the fetus and mother is good. However, physicians need to monitor the evolution of pregnancy more closely. These patients require a caesarian section more frequently than women with a normal pregnancy. This necessity could be explained by the development of placental disease secondary to the underlying disease that led to BCS [103].

## 10. Cirrhosis

Liver cirrhosis is a chronic condition characterized by liver fibrosis and the development of regenerative nodules [111]. The association between liver cirrhosis and pregnancy is very rare. Liver cirrhosis occurs after many years of evolution of the liver disease, with an incidence of only 45 cases per 100,000 women of childbearing age; also, liver cirrhosis induces significant hormonal and metabolic changes, with anovulation, amenorrhea, and reduced fertility [112]. However, advances in the therapeutic management of liver diseases have increased the chances of pregnancy in women with cirrhosis [111,113]. A study conducted between 1993 and 2005, that looked at 339 pregnant women with a positive diagnosis of liver cirrhosis, reported an increase in the number of births from 69 (1993–1999) to 106 (2000–2005) [114]. Another study, published in 2020, reported an increase in childbirth incidence from 2 to 14.9 births per 100,000 subjects between the years 2000 and 2016 [108]. The most common cause of liver cirrhosis in pregnant women is NAFLD [115]. A 2021 study reported viral hepatitis as the most common cause of liver cirrhosis in pregnant women. The same study reported a better prognosis of liver cirrhosis secondary to Wilson’s disease than secondary to hepatitis B virus-induced chronic hepatitis [116].

The diagnosis of liver cirrhosis in pregnant women is similar to the diagnosis in the antepartum period and is based on clinical and paraclinical evaluation (Table 10) [113].

Because of the increasing possibility of associating the two conditions, the risk of maternal–fetal complications remains very high. They fall into two broad categories (Figure 5) [112,114,115].

A study published in 2021 validated the results of another study published in 2018, namely, the absence of significant differences in the rate of miscarriages in women with a positive diagnosis of liver cirrhosis compared to pregnant women without liver cirrhosis [116,117]. The same study reported an increase in blood volume by 30–50% during pregnancy and decreased serum albumin concentration [116]. Increased blood volume can worsen portal hypertension, with high rates of variceal hemorrhage [114]. The compression induced by the pregnant uterus, associated with Valsalva maneuvers, may increase the risk of bleeding during labor [114]. Women who have liver cirrhosis and portal hypertension require a superior digestive endoscopy before pregnancy to identify esophageal varices [118]. In its absence during pre-pregnancy, the endoscopy should be performed in the second trimester. The presence of esophageal varices requires primary prophylaxis with nonselective beta-blockers (preferably propranolol due to its longer half-life) or even endoscopic variceal ligation [118]. Prophylaxis with non-selective beta-blockers should continue during pregnancy, accompanied by careful monitoring of the fetus, due to risks of bradycardia and hypoglycemia [116,118].

With the development of therapeutic options for liver cirrhosis, maternal death rates have decreased from 10% to 1–1.8% [49,119]. One of the main complications of liver cirrhosis is variceal hemorrhage. Treatment with spironolactone should be discontinued. In the case of an acute episode of variceal hemorrhage, therapeutic management involves resuscitation and hemodynamic stabilization of the mother, endoscopic therapy (endoscopic variceal band ligation), and antibiotic prophylaxis [6]. Unlike the antepartum period, the use of vasopressin is not recommended during pregnancy, and the use of octreotide is controversial. When endoscopic therapy has failed, a life-saving therapy may be represented by inserting a transjugular intrahepatic portosystemic shunt [6].

In 2017, Palatnik et al. reported an obstetric complication rate of 61% in women with liver cirrhosis, compared to 12% in the group of women without this chronic liver disease [117]. Newborns from mothers with compensated liver cirrhosis were more prone to death at birth, to be large for gestational age, to have developing respiratory diseases, or even death in the first year of life [114].

As for prognosis, termination of pregnancy in the first trimester has led to avoiding many of the severe complications [116]. Murthy et al. revealed higher rates of Cesarean section in women with decompensated cirrhosis and prior liver transplant. Moreover, clinically apparent decompensated cirrhosis was associated with an increased rate of placenta previa, peripartum blood transfusion, and preterm labor than patients with compensated cirrhosis [120].

## 11. Discussions

In the literature, there is a variability of the real prevalence of liver dysfunction in pregnancy; for each condition, there are differences in diagnosis, treatment, and prognosis according to geographic distribution.

In liver disease (especially in HG), the severity of symptoms (nausea and vomiting) is related to high levels of liver enzymes [7]. Hypertension is more common in HELLP syndrome than in AFLP [47]. The correct diagnosis requires laboratory tests and imaging evaluations, which, unfortunately, in many cases do not provide information about the onset of the disease. The presence of coagulopathy, elevated serum creatinine, and severe hypoglycemia are more frequent features of AFLP than HELLP syndrome. Establishing a differential diagnosis between HELLP and AFLP is difficult due to the large overlap between these two disorders, with delayed diagnosis and adverse consequences on the fetus. Instead, the early establishment of the treatment will determine a good maternal–fetal prognosis. According to the Tennessee and Mississippi classifications, predictable diagnostic criteria are used to manage HELLP syndrome and the Swansea classification for AFLP. The ultrasound reveals no obstruction in HG, hepatic hematomas, or infarcts in HELLP, bright liver in AFLP, and gallstones in cholelithiasis. Furthermore, CT or MRI are useful for excluding the hepatic complications of HELLP syndrome and AFLP and diagnosing BCS.

Subsequent studies have tried to find some specific biomarkers, which are useful for the early diagnosis of these conditions and the assessment of therapeutic efficiency. HELLP syndrome is associated with increased circulating levels of Galectin-1 (gal-1) and p65/RelA immunoexpression of placental NF-kB [15,16] (Table 2). In ICP, the serum levels of the three long noncoding RNA, the maternal urinary mRNAs, or native thiol may improve the diagnosis. Furthermore, increased mean platelet volume (MPV) and total bilirubin levels are associated with preterm delivery. Increased postprandial total serum biliary acid (SBA) levels are predictive for low APGAR in ICP patients. Up-regulation of CRH expression was used in treatment with UDCA in patients with ICP. The diagnosis of certain forms of hepatic impairment is a challenge due to either incomplete diagnostic criteria or specific criteria that overlap in several clinical conditions [17,18,19,20,21,22] (Table 3).

The therapeutic management is specific for various forms of liver disease, from rehydration and antiemetic drugs in HG, rapid delivery with plasmapheresis in AFLP, administration of antibiotics with ERCP or laparoscopic cholecystectomy (I/II trimester) in cholelithiasis, prophylaxis of variceal hemorrhage with endoscopic band ligation in BCS, to active management of varices, correction of coagulopathy and prophylactic antibiotics in cirrhosis. In HELLP syndrome, the therapeutic management is rapid delivery with corticoids. Moreover, corticosteroids decrease the mean arterial pressure in postpartum, reduce the AST levels, and have no effect on clinical outcomes in class I HELLP syndrome [23,24,25,26]. In ICP, UDCA improves maternal pruritus, liver function tests and has no adverse effects on newborns. UDCA monotherapy should be used as the first-line therapy, but also S-adenosylmethionine (SAMe) was effective [27,28,29,30,31,32,33,34,35].

Table 4 summarizes the main RCT studies of treatment in HELLP syndrome and ICP, a permanent concern in establishing the proper therapeutic management of these liver diseases, and Table 5 shows a synopsis of the main therapeutic key points.

The complications of various liver diseases in pregnancy can be misdiagnosed because of the obstetricians’ lack of experience, especially when symptoms are nonspecific. With careful monitoring, the severe evolution of the diseases can be detected early and effectively managed to produce a good prognosis. In many cases, the experience is limited due to the low incidence of liver pathology. The management of these cases must be frequently performed in tertiary centers for high-risk pregnancy by multidisciplinary teams.

More specifically, patients with severe preeclampsia with a significant increase in aminotransferases and a low number of platelets, who do not fall into classical criteria (standard classifications), are diagnosed with incomplete or slowly evolving HELLP syndrome. The differentiation between hepatic dysfunction represents another challenge in preeclampsia from the incomplete form of HELLP or between the severe form of HELLP from acute fatty liver. A mild to moderate form of hepatic dysfunction in pre-eclamptic patients may be present in the first situation. Early diagnosis and identification of risk factors or associated comorbidities can be key points in establishing the appropriate treatment. In contrast, an early diagnosis can positively influence morbidity or maternal–fetal mortality in severe forms of hepatic impairment. Frequently, the fetuses of these mothers with hepatic dysfunction, especially associated with preeclampsia, have severe intrauterine growth restriction, and the rapid evolution of maternal disease can cause stillbirth. Intrauterine deterioration of the fetus is rapidly progressive with the decompensation of maternal liver function, the fetal mortality rate being higher than maternal.

In pregnancy, physiological changes in liver function may worsen the pre-existing liver disease (hepatitis B, C, and autoimmune); diagnosis, monitoring, and treatment are a concern for both the pregnant woman and the obstetrician. Misdiagnosis or non-recognition of these liver diseases can be aggravating elements of the overall prognosis. The management of the pre-existing liver disease requires a follow-up schedule of the pregnant woman in collaboration with a hepatologist or gastroenterologist to have a better maternal–fetal prognosis.

The risk of preterm birth is common among most liver diseases (HELLP syndrome, AFLP, ICP, and cholelithiasis). In the scenario of pregnancies ≥ 23 and <34 weeks of gestation, corticosteroids before delivery are considered safe. Except for the rapid aggravation of the liver disease with reassuring maternal and fetal status before the therapeutic decision, delivery should be considered after 24–48 h from the disease onset.

The maternal–fetal prognosis is good when the diagnosis is established early, comorbidities are identified, a multidisciplinary team makes the therapeutic decision, and the pregnant woman is admitted to a tertiary center specialized in high-risk pregnancy.

The limitations of this review consist of searching for data in only two major databases (PubMed, Web of Science), and we identify CT and RCT for only some clinical conditions (especially for intrahepatic cholestasis and HELLP syndrome).

## 12. Conclusions

Liver disease in pregnancy is a major public health problem associated with increased costs because of the complications that may occur. More epidemiological studies on its prevalence may improve the diagnosis, treatment, and prognosis. Future directions should be more RCT to evaluate the safety and efficacy of treatment for each form of liver dysfunction in pregnancy, to establish new biomarkers for an earlier diagnosis and monitoring therapy, and to make new recommendations to improve perinatal outcomes.

Liver disease during pregnancy is associated with a very high risk of maternal–fetal complications. Careful monitoring of the pregnant woman, early identification of changes in liver function, and the approach by a multidisciplinary team for a rapid and correct diagnosis and individualized treatment decrease the rate of materno–fetal morbidity and mortality.

## Figures and Tables

**Figure 1 healthcare-09-01481-f001:**
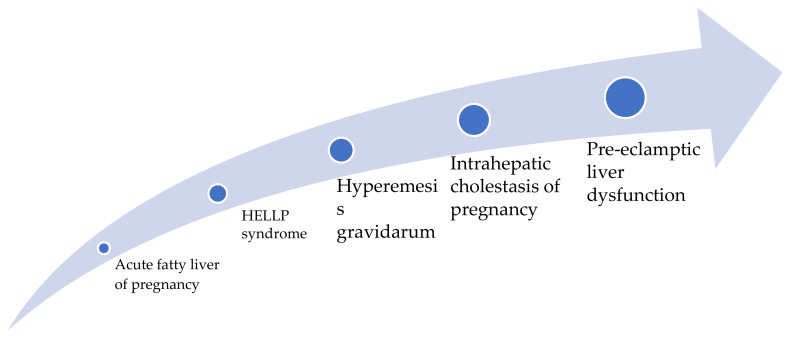
The direction of increased incidence of pregnancy-related liver disease. Data from Ref [7].

**Figure 2 healthcare-09-01481-f002:**
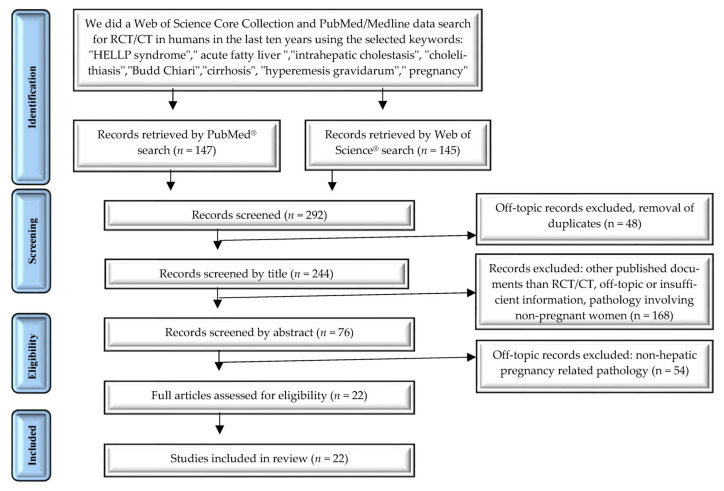
PRISMA diagram: systematic search and study selection process.

**Figure 3 healthcare-09-01481-f003:**
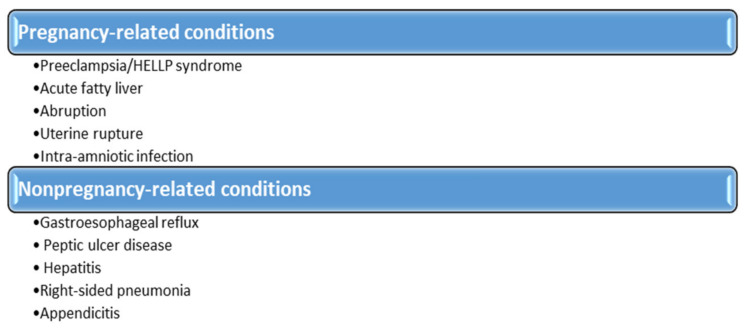
Differential diagnosis of cholelithiasis.

**Figure 4 healthcare-09-01481-f004:**
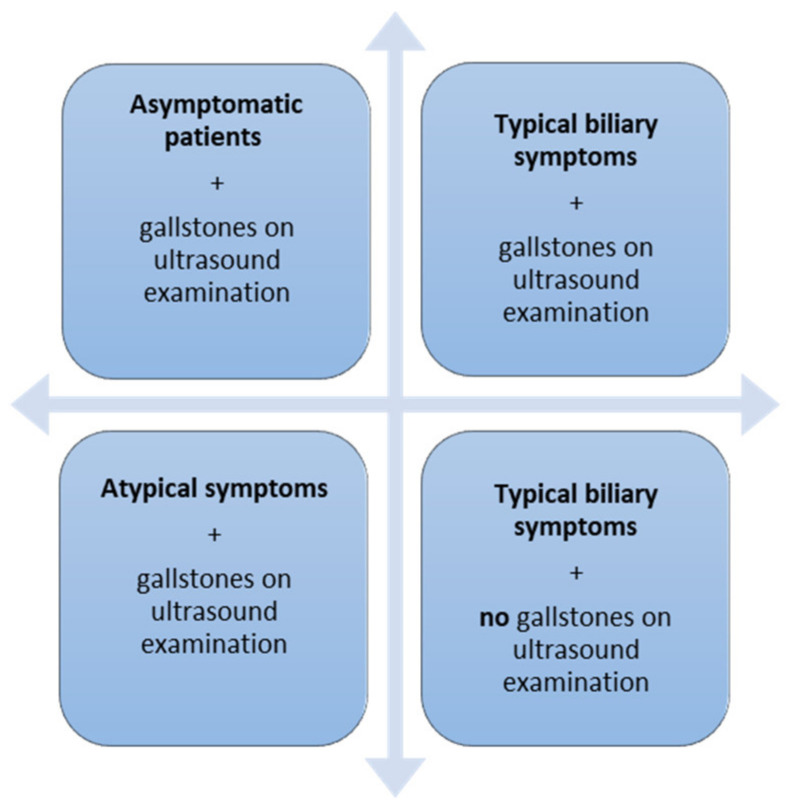
Presentation of women patients with cholelithiasis.

**Figure 5 healthcare-09-01481-f005:**
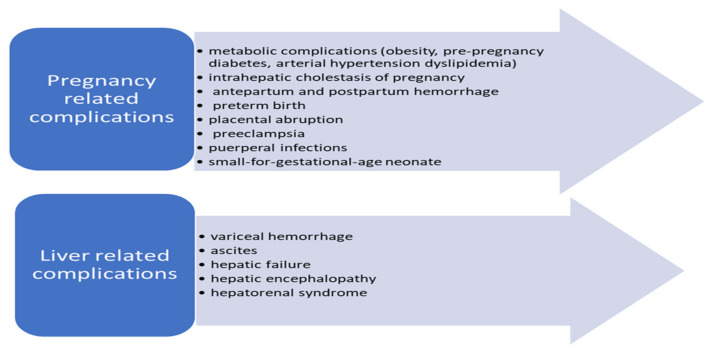
Complications that could develop in pregnant women secondary to liver cirrhosis.

**Table 1 healthcare-09-01481-t001:** Pregnancy-related and unrelated liver disease. Data from Ref [7].

Pregnancy-Related Liver Disease	Pregnancy Unrelated Liver Disease	
	De Novo	Pre-Existing
Pre-eclamptic liver dysfunction	Hepatitis	Acute viral hepatitis
Intrahepatic cholestasis of pregnancy	Cirrhosis	Cholelithiasis
Hyperemesis gravidarum	Autoimmune liver disease	Budd–Chiari syndrome
HELLP syndrome	Wilson’s disease	Metabolic disease
Acute fatty liver of pregnancy	Post liver transplantation	Liver tumors
	Non-alcoholic fatty liver disease	Drug-induced hepatotoxicity

**Table 2 healthcare-09-01481-t002:** Laboratory markers and genetic biomarkers of HELLP. Control = C, Late-onset HELLP *; Early-onset HELLP **; HELLP = H. MPO–myeloperoxidase; CRP–C-reactive protein.

Authors, Publ. Year	Study Design	Cases	Maternal Age	Gestational Weeks	Platelets	Proteinuria	ALT	AST	Biomarker	Conclusion
Simsek, 2013 [15]	CCS	50 20 (H) 30 (C)	28.6 ± 6.8 (H) 28.3 ± 4.8 (C)	33.4 ± 4.7 (H) 37.9 ± 1.2 (C)	91 ± 24.2 (H) 217 ± 68 (C)	NR	248.3 ± 296.8 (H) 12.5 ± 4.5 (C)	352.5 ± 514 (H) 17.7 ± 4.3 (C)	Levels of p65/RelA expression of nuclear transcription factor-kappa beta (NF-kB) in paraffin-embedded placental tissue samples	p65/RelA immunoexpression and serum MPO and CRP levels were significantly higher in patients with HELLP; over-expression of placental NF-kB is correlated with elevation of serum inflammatory markers and placental ultrastructural changes
Schnabel, 2016 [16]	CCS	107; 21 (H) 86 (C)	34.3 ± 4.6 (*) 32.7 ± 4.0 (**) 32.6 ± 5.2 (C)	34–41 (*); 26–33 (**)	130,14 ± 65,08 (*); 109,85 ± 51,67 (**); 196.68 ± 61.60 (C)	4.23 ± 5.18 (*); 4.24 ± 5.03(**); 0.36 ± 0.47 (C)	140.0 ± 200.7 (*); 306.3 ± 213.7 (**); 45.03 ± 117.1 (C)	176.2 ± 308.9 (*); 269.0 ± 268.2 (**); 68.56 ± 159.52 (C)	Galectin-1 (gal-1)	Increased circulating levels of gal-1 are found in HELLP syndrome

**Table 3 healthcare-09-01481-t003:** Laboratory markers and genetic biomarkers of ICP. * ICP, ** Control, CCS = case control study; ICP = intrahepatic cholestasis of pregnancy; NR = not reported; NI = no information; TBA = total bile acids; OS = oxidative stress; LncRNAs = long noncoding RNAs; MPV = mean platelet volume; SBA = serum bile acids; CRH = corticotropin-releasing hormone; UDCA = ursodeoxycholic acid.

Authors, Publ. Year	Study Design	Cases	Maternal Age	Gestational Weeks	Total Bile Acids	Total Bilirubin	ALT	AST	Biomarker	Conclusion
Zhou, 2014 [17]	CCS	30; 16 * 14 **	NR	34 weeks–34 weeks and 6 days;	NR	NR	NR	NR	CRH expression in patients with ICP after UDCA	Maternal serum and placental CRH expression in ICP patients were up-regulated after treatment of UDCA
Ozler, 2014 [18]	CCS	60; 30 * 30 **	29.7 ± 5.9* 29.9 ± 5.0 **	32.8 ± 3.1 * 31.6 ± 4.5 **	NR	NR	242.9 ± 237.8 * 14.9 ± 6.1 **	161.6 ± 140.9 * 12.8 ± 4.1 **	IL-6, TNF-α and neopterin	There was no difference between the groups in IL-6 and TNF-α levels, but the mean neopterin level was significantly higher in group *
Oztas, 2015 [19]	CCS	217; 117 *; 100 **	28 *; 27 **	34.7 ± 2.5 * 34.8 ± 2.5 **	NR	0.7 * 0.47 **	88 * 11.7 **	64 * 16.8 **	Mean platelet volume, total bilirubin levels, increased postprandial total SBA levels	Increased MPV and total bilirubin levels are associated with preterm delivery, and increased postprandial total SBA levels are predictive for low APGAR in ICP patients
Ma, 2016 [20]	CCS	90; 40 * 50 **	28.8 ± 4.1 * 28.9 ± 3.4 **	34.0 ± 2.1 * 32.3 ± 2.5 **	26.7 ± 21.1 * 2.6 ± 1.3 **	10.4 ± 5.6 * 7.6 ± 2.6 **	88.7 ± 99.7 * 15.9 ± 17.7 **	53.1 ± 49.2 * 18.7 ± 9.6 **	Urinary miRNAs as non-invasive biomarkers for ICP	Urinary microRNA profiling detection is feasible and has the potential to be noninvasive biomarkers for the diagnosis of ICP
Sanhal, 2018 [21]	CCS	107; 57 * 50 **	27.9 ± 5.3 * 27.3 ± 5.6 **	35 * 35 **	NR	0.5 * 0.37 **	92 * 9 **	66 * 16 **	Thiol/disulfide to evaluate oxidative stress (OS).	Pregnant women with ICP had significantly lower serum levels of native total thiol and higher levels of disulfide; thiol/disulfide balance indicate OS in the pregnant woman with ICP; favorable diagnostic abilities of native thiol and total thiol in ICP
Zou, 2021 [22]	CCS	108	28.9 ± 6.32 *; 26.1 ± 4.87 **	37.9 ± 0.9 * 38.9 ± 0.9 **	68.9 ± 50.3 * 5.3 ± 3.1 **	NR	108.6 ± 101.4 * 16.1 ± 7.6 **	106.7 ± 96.1 * 16.9 ± 8.3 **	Long noncoding RNAs (lncRNAs)	The three lncRNAs serum level are potential biomarkers of ICP. Combining with TBA, alanine aminotransferase, and glycocholic acid, may improve the diagnosis of ICP.

**Table 4 healthcare-09-01481-t004:** Main RCT/CT studies for treatment in HELLP, ICP and AFLP. I = group A, II = group B. PPC-Polyunsaturated phosphatidylcholine; SAMe-S-adenosylmethionine.

Authors, Publ. Year	Study Design	Pathology	Cases	Maternal Age	Gestational Weeks	Total Bile Acids	Direct Bilirubin	ALT (Mean ± SD)	AST (Mean ± SD)	Treatment	Results
Katz, 2013 [23]	Triple blind RCT	HELLP class I syndrome	400	NI	NI	NI	NI	NI	NI	Eligible patients receive dexamethasone every 12 h for two days	Corticosteroids increase platelet counts significantly, with no clear evidence of the effect on clinical outcomes
Shahzad, 2017 [24]	RCT	HELLP syndrome	100	30.5 ± 5.8	39.6 ± 1.1	NI	NI	NI	NI	Group A: 10 mg dexamethasone sodium phosphate IV every 12 h; Group B: 12 mg combination of betamethasone acetate and betamethasone sodium phosphate IM every 24 h	Decrease in mean arterial pressure with dexamethasone was significantly higher than that of betamethasone for management of females presenting with postpartum HELLP syndrome
Fonseca, 2019 [25]	RCT	HELLP class I syndrome	87	25.7 ± 7.5	33.8 ± 4.8	NR	NR	188.4	337.4	Pregnant women: 10 mg doses of dexamethasone sodium phosphate, IV, every 12 h until delivery; and 3 additional doses after delivery. Postpartum women: three 10-mg doses after delivery.	Failed to demonstrate the benefit of using dexamethasone in patients with class I HELLP syndrome
Takahashi, 2019 [26]	Retro- spective study	HELLP syndrome	18 7 * 11 **	30.2 ± 4.3 * 30.3± 5.3 **	36.5 ± 5.3 * 37.1± 4.5 **	NR	NR	NR	177 ± 128 * 399± 228 **	Group *: without dexamethasone Group **: with IV dexamethasone	AST levels were significantly higher in group *. No maternal postpartum complications between the groups.
Marciniak, 2011 [27]	CCS	ICP	43	NI	NI	NI	NI	NI	NI	Group 1: PPC Group 2: UDCA Group 3: a combination of these two drugs	Combined therapy with UDCA and PPC could be considered in ICP, especially in case of early-onset and/or severe course
Chappell, 2012 [28]	Semi-factorial RCT	ICP	111	29.8 ± 5.7	34.2 ± 3	25.95	9	94	59.7	UDCA (250 mg dose) or placebo capsules, two capsules twice a day, and if there was no improvement, the dose was increased in increments of two capsules per day every 3–14 days up to a maximum of 2 g/day	UDCA significantly reduces pruritus, but the size of the benefit may be too small for most doctors to recommend it, or for most women to want to take it
Joutsiniemi, 2013 [29]	Double blind RCT	ICP	20	NI	32.6	NI	NI	NI	NI	Random administration of 450 mg/day UDCA or placebo for a period of 14 days during the third trimester of pregnancy	UDCA significantly improves maternal pruritus, improves liver function tests and has no adverse effects on newborns
Jain, 2013 [30]	Prospective randomized study	ICP	69	27.5 ± 4.3	35	NI	NI	165.6 ± 116.4	145.9 ± 102.6	Group I was planned for delivery at 37 weeks. In Group II, pregnancy was carried to 38 weeks under surveillance. Fetal surveillance start at >34 weeks at diagnosis and included daily maternal records of fetal movements, biophysical profiles. Fetal monitoring was conducted weekly before 36 weeks and biweekly after that	With active intervention, pregnancies with obstetric cholestasis can be carried to a later gestation under surveillance
Zhang, 2015 [31]	RCT	ICP	120	28.2 ± 3.9	31.1 ± 3.3	44.2 ± 40.3	22.4 ± 18.7	259.3 ± 173.7	187.5 ± 124.7	Group 1: oral UDCA 4×250 mg daily until delivery. Group 2: IV SAMe 1000 mg daily until delivery. Group 3:a combination of these two drugs in the same dosage until delivery	UDCA and SAMe are safe and effective in ICP treatment. UDCA monotherapy should be used as the first-line therapy because it is more efficacious, cost-effective and convenient
Grymowicz, 2016 [32]	CCS	ICP	303; 203 *: I = 46 (TBA < 10 mmol/l); II = 157 (TBA > 10 mmol/l) 100 **	NR	34.4 ± 3.4	9.5 (I) 21.8 (II)	14.2 (I) 14.8 (II)	158.96 (I) 214.36 (II)	105.62 (I) 138.34 (II)	Only group A: UDCA (300–450 mg/day; 4–6 mg/kg/day) until delivery	Low doses of UDCA improved clinical symptoms and biochemical markers in almost 90% of patients
Parízek, 2016 [33]	Retro- spective multicentric study	ICP	191	31.9 ± 4.6	37.4	20.5	NR	237 ± 204	145 ± 120	UDCA was used in the range of 500–1500 mg/day, 750 mg/day in most cases (10 mg/kg/day). The average duration of therapy was 17 days	UDCA ameliorated liver dysfunction in the majority of the affected women (86.1%)
Chappell, 2019 [34]	RCT	ICP	604	30.6 ± 5.4	34.4	27.5	8.23	64.75	55.42	UDCA or placebo, given as two oral tablets a day at an equivalent dose of 500 mg twice a day (maximum of 4 tablets and a minimum of one tablet a day) from enrolment until the infant’s birth	Treatment with UDCA does not reduce adverse perinatal outcomes in women with ICP
Agarwal, 2021 [35]	Prospectivestudy	ICP	121 71 ^A^ 50 ^B^	27.7 ± 3.8 ^A^ 27.2 ± 3.5 ^B^	30.4 ± 4.7 ^A^ 31.2 ± 3.3 ^B^	75.9 ± 39.5 29.2 ± 5.7	173.3 ± 139 28.9 ± 8.2	NR	173.3 ± 139 28.9 ± 8.2	Group ^A^-oral UDCA 300 mg thrice daily	In Asian/Indian patients, biliary acids (BA) levels are higher compared to the general population. In this case, it’s necessary to establish a higher BA cut-off of 30 μmol/L for diagnosing ICP.
Chu, 2012 [36]	CCS	AFLP	11	26 ± 4.2	33	-	-	-	-	Plasma exchange and continuous hemodiafiltration were used in ten patients who were cured and discharged from the hospital. The hospitalization average duration was 17 days	This is an effective treatment for patients with AFLP suffering multiple organ dysfunction.

Group *: without dexamethasone, Group **: with IV dexamethasone; Group A: ICP patient, Group B: pregnant patient without ICP; h-hours.

**Table 5 healthcare-09-01481-t005:** Synopsis across the main findings in hepatic dysfunction in pregnancy. ERCP: endoscopic retrograde cholangiopancreatography; DIC-: disseminated intravascular coagulation; NICU: neonatal intensive care unit; MELD score: model for end-stage liver disease.

Features	Hyperemesis Gravidarum	HELLP Syndrome	Acute Fatty Liver (AFLP)	Intrahepatic Cholestasis (ICP)	Cholelithiasis	Budd–Chiari Syndrome (BCS)	Cirrhosis
Epidemiology	0.3–2%	0.1%–0.6%	0.01–0.02%	0.2–2% with seasonal pattern	5–12% no symptom gallstones Gallbladder disease 0.05–0.3%	0–21.5%	0.045%
Moment of appearance	First trimester	Late second trimester (25–38 weeks) to early postpartum	Third trimester (32–38 weeks) –postpartum	Second or third trimester (21–38 weeks)	-	-	-
Clinical findings	Intense nausea, vomiting, nutritional deficiency, weight loss	Abdominal pain, nausea/vomiting, overlap with preeclampsia, hypertension, and proteinuria	Abdominal pain, nausea/vomiting, malaise, anorexia, jaundice, hypoglycemia, signs of liver failure, ascites	Pruritus, dark urine, jaundice, loss of appetite, fatigue, nausea, steatorrhea, abdominal pain	Stabbing pain or colicky in the right upper quadrant and/or epigastric area, anorexia, nausea, vomiting, dyspepsia, low-grade fever, tachycardia, and fatty food intolerance	Abdominal pain, hepatomegaly, ascites, variceal bleeding	Variceal bleeding (20%–25%), especially during the second trimester or during labor; jaundice, pruritus, nausea, vomiting
Imagistic findings	No biliary obstruction	Hepatic hematomas, infarcts, possible rupture	Bright liver secondary liver fatty infiltration	Exclusion diagnosis with cholelithiasis	Ultrasound 95% effective	MRI and ultrasound are most effective	Endoscopy: esophageal and gastric varices
Histologic findings	-	Variable periportal necrosis	Microvesicular fatty infiltration	Dilated bile canaliculi	Biliary sludge-plate-like cholesterol crystals and calcium bilirubinate granules embedded in strands of mucin gel	Zone 3 hemorrhage into liver cell plates and ischemic necrosis leading to veno-centric cirrhosis	Diffuse disruption in the architecture of the entire liver (loss of the normal central–portal relationship)
Laboratory findings	↑ALT 1–2 x (50%)	↑ALT up to 2–30 fold ↑Total bilirubin up to 1.5–10 fold ↓platelets LDH >600 IU/mL	↑ALT up to 3–15 fold ↑bilirubin up to 3–15 fold, ↑uric acid (>340 μmol/L), ↓glucose; ↓antithrombin III creatinine > 150 μmol/L; DIC; ↓platelets	↑ALT up to 2–10 fold ↑total bile acid up 10–100 fold; ↑GGT up to 0–4 fold; ↑Alkaline phosphatase up to 7–10 fold	↑bilirubin ↑ALT ↑AST	Hypercoagulability blood volume expansion and hypoproteinemia	AST and ALT usually moderately elevated Other parameters depend on etiology
Therapeutic management	Rehydratation, antiemetic drugs, vitamins (C, B1, B6, B12)	Rapid delivery (34 weeks or before 24–34 weeks with corticoids for fetal lung maturation	Rapid delivery, plasmapheresis, liver transplantation	UDCA (10–20 mg/kg/day)	Discontinuation of oral intake, IV fluid replacement, analgesia, and administration of antibiotics when signs of infection are present laparoscopic cholecystectomy (I/II trimester) or ERCP	Prophylaxis of variceal hemorrhage; large or ‘at-risk’ varices should be eradicated with endoscopic band ligation. Preferable mode of delivery: assisted vaginal delivery with adequate analgesia:; Caesarean section reserved for obstetric indications	Active management of varices. Preferable vaginal deliveries. Cesarean in case of large varices. Correction of coagulopathy and prophylactic antibiotics to reduce postpartum hemorrhage and bacterial infections.
Prognosis	Remission in the first part of the second trimester (18 wks)	Usually resolves by the first part of the second trimester	Small risk of recurrence; maternal mortality has decreased to <10% in most recent series	Good after delivery	Good	Favorable in patients with treated and stabilized BCS	Increased maternal and fetal problems
Fetal outcome	Not associated with adverse pregnancy outcomes	Fetal bradycardia, fetal loss, fetal distress, Premature birth	Premature birth, newborn asphyxia, mortality rate: 23%	Preterm birth, sudden intra-uterine death, meconium-stained amniotic fluid, NICU admission	Risk of preterm birth and neonatal morbidity	Fetal outcomes beyond 20 weeks gestation are good	Related to the severity of the maternal liver disease (MELD score: the risk of decompensating of the maternal liver)

**Table 6 healthcare-09-01481-t006:** Forms of HELLP syndrome according to Mississippi and Tennessee classifications. Data from Ref. [48].

HELLP Class	Mississippi Classification		
	Platelet Count	AST/ALT	LDH
1–mild	100,000–150,000/L	>40 IU/L	>600 IU/L
2–moderate	50,000–100,000/L	>70 IU/L	
3-severe	<50,000/L	>70 IU/L	
	HELLP class	Tennessee classification	
Severe preeclampsia	Complete (3 criteria)	Platelet count <100 × 10^9^/L	
		AST >70 IU/L and LDH >600 IU/L	
		Bilirubin ≥1.2 mg/dL	
	Incomplete/partial	One or two 2 criteria	

**Table 7 healthcare-09-01481-t007:** Swansea criteria used in AFLP diagnosis.

Clinical Findings	Laboratory Findings
✓ Nausea	✓ Elevated Bilirubin >0.8 mg/dL
✓ Vomiting	✓ Leucocytosis >11 × 109/L
✓ Abdominal pain	✓ Hypoglycemia <72 mg/dL
✓ Ascites or bright liver by ultrasound	✓ Elevated urea >950 mg/dL
✓ Polydipsia/polyuria	✓ Elevated transaminases AST/ALT >42 U/L
✓ Encephalopathy	✓ Elevated ammonia >66 μmol
✓ Microvesicular steatosis in liver biopsy	✓ Coagulopathy or PT >14 s or aPTT >34 s
✓ Renal impairment	✓ Creatinine >1.7 mg/dL

**Table 8 healthcare-09-01481-t008:** Diagnosis of cholelithiasis in pregnancy. Data from Ref [93,94,95].

Clinical Features	Laboratory Features	Imaging Features
Absence of symptoms Typical biliary symptoms: pain, nausea, vomiting, anorexia. Atypical symptoms: belching, regurgitation, early satiety, abdominal distension, nausea or vomiting alone, epigastric or retrosternal burning, chest pain, or nonspecific abdominal pain.	Absence of complications: laboratory tests are usually normal Presence of complications: leukocytosis, increased liver tests (aminotransferases, bilirubin, alkaline phosphatase), or pancreatic tests (amylase, lipase)	Ultrasonography: gallbladder stones, ultrasonographic Murphy’s sign, gallbladder distension, pericholecystic fluid, wall thickening MRI: helpful when some complications (choledocholithiasis, pancreatitis) are suspected HIDA scan: rarely recommended ERCP: identification of gallstones

**Table 9 healthcare-09-01481-t009:** Diagnosis of BCS in pregnancy. Data from Ref [109].

Clinical Features	Laboratory Features	Imaging Features
‑ Absence of symptoms. ‑ Most common manifestations: fever, abdominal pain, abdominal distention, ascites, lower extremity edema, gastrointestinal bleeding, and encephalopathy.	‑ Liver function tests: normal or slightly modified. ‑ Fulminant liver failure (elevation of hepatic enzymes, coagulopathy) and renal failure.	‑ Conventional and Doppler ultrasound: hepatomegaly, enlarged caudate lobe, splenomegaly, ascites, lack of visualization of the hepatic veins, intrahepatic collaterals, compressed inferior vena cava. ‑ Computed tomography (CT): normal liver morphology, hepatomegaly, enlarged caudate lobe, compressed IVC, absence of hepatic veins, ascites. ‑ Magnetic resonance imaging (MRI): hepatomegaly, enlarged caudate lobe, regenerative nodules, intrahepatic collaterals, absence of blood flow in the occluded veins, detection of thrombus, ascites. ‑ Catheter Venography: the reference standard for diagnosing BCS: anatomic information, hemodynamic information, and histologic information by transjugular liver biopsy.

**Table 10 healthcare-09-01481-t010:** Diagnosis of liver cirrhosis in pregnant women. Data from Ref [113].

Clinical Features	Laboratory Features	Imaging Features
‑ Asymptomatic cirrhosis (incidental discovery) ‑ Presentation with life-threatening complications: variceal hemorrhage, hepatic encephalopathy, ascites, spontaneous bacterial peritonitis.	‑ Macro-, normo- or microcytic anemia; thrombocytopenia; leukopenia; coagulopathy ‑ Elevation of: hepatic enzymes, bilirubin, ammonia ‑ Hypoalbuminemia ‑ Hyponatremia	‑ Ultrasonography: nodularity and increased echogenicity of the liver, atrophy of the right lobe, hypertrophy of the left and caudate lobes, evaluation of portal and central vein diameters and velocities ‑ MRI: helpful when hepatocellular carcinoma (HCC) or vascular lesions are suspected; determination of hepatic iron or fat content ‑ Elasticity measurement ‑ Liver biopsy: the gold standard for diagnosis of liver cirrhosis

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
