# Peer review of "State of the Art in Hepatic Dysfunction in Pregnancy"

_healthcare, 2021, doi:10.3390/healthcare9111481_

Round 1

Reviewer 1 Report

Major comments:

1.In this article, the classification of hepatic dysfunction in pregnancy is confused, which should be divided into pregnancy-specific diseases and chronic liver diseases that may be affected or aggravated by pregnancy [PMID:32583511]. Especially in Tab.2, the Mississippi classification and Tennessee classification of HELLP syndrome are not correct. Please make further corrections.

2.There are 20 articles included in this review, and a meta-analysis can be conducted to analyze a specific problem like HELLP syndrome and express it more clearly. And the meta-analysis process is not rigorous, in which the inclusion criteria and exclusion criteria are not clear. It is suggested to elaborate further.

3.The sentence “The authors performed a systematic review by online search compliant to the Preferred Reporting Items for Systematic Review and Meta-Analysis (PRISMA) (Figure 2) guidelines on published articles (Page 3, Line 23-24)” is inaccurate. According to 2020 PRISMA checklist and PRISMA 2020 flow diagram [PMID: 33782057]:

(1) The authors didn't describe the eligibility criteria, including the baseline criteria, inclusion and exclusion criteria for the review, and how studies were grouped for the syntheses, as well as data extraction.

(2) In the results section, the author needs to supplement appraising methodological quality, levels of evidence and present assessments of risk of bias for each included study. An analysis of the results is required, not just to tell how many studies are in this database.

(3) In the screening process, instead of "off-topic records excluded," the authors should explain how the records are excluded.

4.The discussion part is not comprehensive enough to thoroughly discuss the diseases mentioned in the article.For example, the third line of the second paragraph in the discussion section said that “The correct diagnosis requires laboratory tests and imaging evaluations,” but mainly didn’t discuss the tests and evaluations.

Minor comments:

1.In Fig.1, with the direction of the arrow, does the incidence of disease increase gradually? Please further explain the meaning of the figure.

2.It would be better to elaborate on the management of hyperemesis gravidarum. More detailed treatments vary according to their severity, including the presence or absence of hypovolemia, including the use of different medications [PMID:29266076].

3.Tab.3, Tab.4, and Tab.6 are incomplete. Please resize the table.

4.In Section 6, it mentioned that “AFLP is a rare disorder, with an approximate incidence of 1 in 7000 to 1 in 20,000 deliveries.” But no reference was given.

Author Response

Reviewer 1

Dear Esteemed Reviewer,

Thank you for revising our manuscript.

1.In this article, the classification of hepatic dysfunction in pregnancy is confused, which should be divided into pregnancy-specific diseases and chronic liver diseases that may be affected or aggravated by pregnancy [PMID:32583511]. Especially in Tab.2, the Mississippi classification and Tennessee classification of HELLP syndrome are not correct. Please make further corrections.

Answer: Thanks for your remark. We corrected Table 2 in accordance with your reccomandation and we added in the Table 1 a new column to classified hepatic dysfunction in pregnancy (please see the attached manuscript).

2.There are 20 articles included in this review, and a meta-analysis can be conducted to analyze a specific problem like HELLP syndrome and express it more clearly. And the meta-analysis process is not rigorous, in which the inclusion criteria and exclusion criteria are not clear. It is suggested to elaborate further.

Answer: Thanks for your mention about this situation. We corrected in accordance with your reccomandation and we introduce the inclusion and exclusion criteria in study design. (please see the attached manuscript – 2.2 section)

3.The sentence “The authors performed a systematic review by online search compliant to the Preferred Reporting Items for Systematic Review and Meta-Analysis (PRISMA) (Figure 2) guidelines on published articles (Page 3, Line 23-24)” is inaccurate. According to 2020 PRISMA checklist and PRISMA 2020 flow diagram [PMID: 33782057]:

Answer: We corrected in accordance with 2020 PRISMA checklist. (please see the attached manuscript – 2.2 section)

 (1) The authors didn't describe the eligibility criteria, including the baseline criteria, inclusion and exclusion criteria for the review, and how studies were grouped for the syntheses, as well as data extraction.

Answer: Thanks for your remark. We introduce in 2.2. and 2.3 sections the inclusion and exclusion criteria, and data extraction and synthesis. (please see the attached manuscript – 2.2 and 2.3 sections)

 (2) In the results section, the author needs to supplement appraising methodological quality, levels of evidence and present assessments of risk of bias for each included study. An analysis of the results is required, not just to tell how many studies are in this database.

Answer: Thanks for your mention, but we didn’t perform a meta-analysis, because there is a small number of clinical trials and randomized clinical trials. (please see the attached manuscript)

 (3) In the screening process, instead of "off-topic records excluded," the authors should explain how the records are excluded.

Answer: Thanks for your indication. We explain the reasons of exclusion of the records (Figure 2) and 3.1 section. (please see the attached manuscript)

4.The discussion part is not comprehensive enough to thoroughly discuss the diseases mentioned in the article. For example, the third line of the second paragraph in the discussion section said that “The correct diagnosis requires laboratory tests and imaging evaluations,” but mainly didn’t discuss the tests and evaluations.

Answer: Thanks for your remark. We made tables in each section to summarize the main clinical manifestations, laboratory and imaging findings, and created a synopsis of the main liver diseases in pregnancy in discussion section which includes the diagnosis and therapeutic decisions (see Table 6). (please see the attached manuscript)

Minor comments:

1.In Fig.1, with the direction of the arrow, does the incidence of disease increase gradually? Please further explain the meaning of the figure.

Answer: Thanks for your mention, we corrected for a better understanding of the increasing incidence of pregnancy-related liver disease (please see the attached manuscript)

2.It would be better to elaborate on the management of hyperemesis gravidarum. More detailed treatments vary according to their severity, including the presence or absence of hypovolemia, including the use of different medications [PMID:29266076].

Answer: Thank you for your mention, we detailed the therapeutic decisions of hyperemesis gravidarum. Also, we introduced new recent references. (please see the attached manuscript).

3.Tab.3, Tab.4, and Tab.6 are incomplete. Please resize the table.

Answer: Thank you for your remark, but it was a problem with the resize of the tables, because we made them in landscape format. (please see the attached manuscript).

4.In Section 6, it mentioned that “AFLP is a rare disorder, with an approximate incidence of 1 in 7000 to 1 in 20,000 deliveries.” But no reference was given.

Answer: Thank you for your mention, we introduced new recent references. (please see the attached manuscript).

Reviewer 2 Report

Tables 3,4,6, are not fully displayed in the PDF file you sent me.

This limitation (that I communicated to the Editor by an email 19-08-21) made difficult to me the review process.

The order of the page numbers does not seem correct; for example, after page number 7, there is page 2,3, and so on.

If the research is a Systematic Review, probably it would be better mention it in the abstract too. 

Pag. 4

Results

I would suggest to move this paragraph in the Material and Methods Section.

In fact, I believe that the reader would expect finding Results Section beginning with the  paragraph

Hyperemesis gravidarum and liver dysfunction.

Pag. 7

HELLP syndrome, 18% risk of preeclampsia, and a 20% risk of gestational hypertension).

Please, complete the bracket.

Pag. 12

The presence of oesophageal varices require primary prophylaxis with nonselective beta-blockers…

In pregnant women with esophageal varices, prophylaxis with beta-blockers (propranolol or nadolol) should be initiated..

Avoid repetitions!

C-section.

It is better specifying Caesarean section.

Pag. 13

plasmapheresis

this kind of treatment the Authors mention in the Discussion is not mentioned in the results but only in the table 5.

Specify all the acronyms, in the text and in the tables.

Be sure that in the References are cited the 22 articles you focused on for your systematic review.

It is my impression that you have performed the systematic review only for some but not all the clinical conditions you have covered in the Results.

If so, maybe you should add this limitation at the end of your paper.

Author Response

Reviewer 2

Dear Esteemed Reviewer,

Thank you for revising our manuscript.

Tables 3,4,6, are not fully displayed in the PDF file you sent me. This limitation (that I communicated to the Editor by an email 19-08-21) made difficult to me the review process.

The order of the page numbers does not seem correct; for example, after page number 7, there is page 2,3, and so on.

Answer: Thank you for your remark, but it was a problem with the resize of the tables, because we made them in landscape format. We resize the tables (please see the attached manuscript).

If the research is a Systematic Review, probably it would be better mention it in the abstract too. 

Answer: Thanks for your remark. We introduce this mention in the abstract. (please see the attached manuscript)

Pag. 4

Results

I would suggest to move this paragraph in the Material and Methods Section.

In fact, I believe that the reader would expect finding Results Section beginning with the paragraph

Hyperemesis gravidarum and liver dysfunction.

Answer: Thanks for your mention. We move the paragraph and write study characteristics in the Results section. (please see the attached manuscript – section 3)

Pag. 7

HELLP syndrome, 18% risk of preeclampsia, and a 20% risk of gestational hypertension).

Please, complete the bracket.

Answer: Thanks for your remark. We correct and complete the bracket. (please see the attached manuscript)

Pag. 12

The presence of oesophageal varices require primary prophylaxis with nonselective beta-blockers…

In pregnant women with esophageal varices, prophylaxis with beta-blockers (propranolol or nadolol) should be initiated..

Avoid repetitions!

C-section. 

It is better specifying Caesarean section.

Answer: Thanks for your mention. We keep only one sentence and correct C-section in the whole document. (please see the attached manuscript)

Pag. 13

plasmapheresis

this kind of treatment the Authors mention in the Discussion is not mentioned in the results but only in the table 5.

Answer: Thanks for your mention. We correct this omission. (please see the attached manuscript)

Specify all the acronyms, in the text and in the tables.

Answer: Thanks for your mention. We specify all the acronyms (please see the attached manuscript).

Be sure that in the References are cited the 22 articles you focused on for your systematic review.

Answer: Thanks for your remark. We added four more references that we discussed in the text. (please see the attached manuscript).

It is my impression that you have performed the systematic review only for some but not all the clinical conditions you have covered in the Results. If so, maybe you should add this limitation at the end of your paper.

Answer: Thanks for your remark. We identify CT and RCT only for some clinical conditions (especially for intrahepatic cholestasis and HELLP syndrome). We add this limitation at the end of the paper in accordance with your suggestion (please see the attached manuscript).

Round 2

Reviewer 1 Report

Major comments:

  1. There have been some reviews published about diagnosing and managing pregnancy-specific liver disorders. The authors should compare the current study with prior publications [PMID:26658682,32583511, 32025601,34602893] in the introduction section to better explain the aim of this study.
  2. Page 4, in the results section, 22 papers were identified, analyzed and included in the review. But only 14 of the included studies were provided. The authors need to explain about it.
  3. In this review, the authors obtained the information about “(1) biomarkers of ICP among studies, (2) biomarkers of HELLP syndrome, (3) treatment options in the most frequent dysfunctions of the liver in pregnancy, (4) main findings in hepatic dysfunction in pregnancy (Page 3, study selection)” by searching the literature, and the references 12-25 were the results shown by the authors. However, these results are barely mentioned in the subsequent discussion about therapeutic decisions of this article. It would be better for the authors to discuss the searching results in more depth.

Minor comments:               

  1. In Fig.1,with the direction of the arrow, the incidence of disease increases gradually? Please further explain the meaning of the figure.
  2. The serial number of Table 2, 3, 4, 5 are not correct, please check it carefully.

Author Response

Reviewer 1

Dear Esteemed Reviewer,

Thank you for revising our manuscript.

Major comments:

  1. There have been some reviews published about diagnosing and managing pregnancy-specific liver disorders. The authors should compare the current study with prior publications [PMID:26658682, 32583511, 32025601, 34602893] in the introduction section to better explain the aim of this study.

Answer: Thanks for your remark. We added references following your recommendation and compared our study with prior publications as you mentioned (please see the attached manuscript).

  1. Page 4, in the results section, 22 papers were identified, analyzed, and included in the review. But only 14 of the included studies were provided. The authors need to explain about it.

Answer: We corrected this and we introduced all the papers in the results section. (please see the attached manuscript).

  1. In this review, the authors obtained the information about “(1) biomarkers of ICP among studies, (2) biomarkers of HELLP syndrome, (3) treatment options in the most frequent dysfunctions of the liver in pregnancy, (4) main findings in hepatic dysfunction in pregnancy (Page 3, study selection)” by searching the literature, and the references 12-25 were the results shown by the authors. However, these results are barely mentioned in the subsequent discussion about therapeutic decisions of this article. It would be better for the authors to discuss the searching results in more depth.

Answer: Thanks for your important indication. We included in the results all 22 papers (references 15 - 36) and mentioned them in tables, in the synopsis, and in the discussions section. (please see the attached manuscript).

Minor comments:                

  1. In Fig.1, with the direction of the arrow, the incidence of disease increases gradually? Please further explain the meaning of the figure.

Answer: Thanks for your indication. We explain the meaning of the figure in the text. (please see the attached manuscript).

  1. The serial number of Table 2, 3, 4, 5 are not correct, please check it carefully.

Answer: Thanks for your remark. We checked and corrected all the tables and put them in the proper position. (please see the attached manuscript).

Kindest regards!

This manuscript is a resubmission of an earlier submission. The following is a list of the peer review reports and author responses from that submission.

Round 1

Reviewer 1 Report

Major comments:

  1. Part of the content in the manuscript have been published[PMID:32583511] .So,this manuscript lack of originality. It is suggested that the HELLP class (mild, moderate and severe) should be presented in the manuscript.
  2. It may be deserved to be considered whether themanuscript can express the clinical features, the laboratory features and imaging features in a table in every subheading paragraph.
  3. This article is aimed at reviewing the diagnosis and the most important therapeutic decisions that a clinician should take in severe hepatic dysfunction during pregnancy, but the drug therapy discussed is limited. The adequate references should be reviewed to enrich the article.
  4. It can be considered that whether the part of discussion can be added into the manuscript.

Minor comments:

     1.It is suggested to review updated references and show more advancement.  

     2.The diseases listed by the authors belong to different classifications and do not demonstrate common characteristics as diseases that may lead to severe hepatic dysfunction. Suggested supplement common characteristics.

Reviewer 2 Report

In the present manuscript, the authors comprehensively summarised the various aspects of liver dysfunction observed during pregnancy, from HELLP syndrome to cholelithiasis to Budd-Chiari syndrome. While not entirely novel, the manuscript represents a massive effort of the authors and is well-written, and my evaluation is that the paper's contents will be of great interest to the readership of Healthcare.I assume, however the manuscript could be more concise to the readers, who are regarded to be general Ob/Gyn doctors and physicians rather tha specialist hepatologists, if the authors provide with when to refer to specialists etc.The manuscript is thorough and fun to read nonetheless.

Reviewer 3 Report

Dear Authors,

I would like to share my comments regarding Your manuscript. The review is carefully written, though my most important concern about the paper is poor novelty of presented review. Most of paragraphs contain well established knowledge. I miss fresh references regarding new research in the field. If I may recommend, it would be interesting to include research about the improvements in the diagnostics, management and therapy in this field. The future directions should be mentioned, as well.  As the ultrasound is so easy accessible nowadays, it would be valuable for clinicians to mention about it in a greater extent in particular pathologies. The discussion about the physiology of hepatic functional changes in pregnancy should be also expanded.

Kind regards,

Reviewer 4 Report

In the Study by Nicolae et. al. the authors present a literature review of hepatic dysfunction in pregnancy. I commend the authors on this interesting topic. The authors have presented a summary of all the conditions with some important clinical pearls. However, I have some concerns about the literature review performed by the authors and the information presented. To make it more interesting and a good read to clinicians from all fields such as primary care physicians, internists, hospitalists and family medicine physicians, I suggest the following changes-

  • In the introduction, the authors could discuss about the prevalence of such across the world, outcomes in pregnancy, and further important characteristics. In the introduction, the authors have cited only 1 article- thereby important information about epidemiology is missing in a review article.
  • Across each syndrome the authors are missing appropriate citations- for example in HELLP syndrome- definition of HELLP syndrome (no citation- named by Dr. Louis Weinstein in 1982), “It can be related to preeclampsia, but in 15-20% of cases, there is no personal history of hypertension or proteinuria” no citation provided. “There are several risk factors for HELLP syndrome (personal history of preeclampsia or HELLP syndrome, or multiparity). The signs and symptoms of HELLP syndrome are less specific and can be easily mistaken for gastroenteritis, gallbladder disease, or viral hepatitis”- No citation provided. HELLP syndrome is considered an obstetric emergency (no citation)- the same is across each condition.
  • Citation numbers 1,2, 13, 17, 27, 38, 48 are uptodate articles- This seriously undermines the study- the authors are trying to present a review article- as a reader, I can easily log in to uptodate to gather information rather than reading the current study. It just seems that the authors have more or less reworded the information and presented a review article, rather than perform a comprehensive literature review.
  • The authors could try and present the information in a small summary table or a diagram for each condition which will be more interesting to the authors.